# Granular analysis of pathways to care and durations of untreated psychosis: A marginal delay model

Walter S. Mathis[1,2] *, Maria Ferrara[1,2], Shadie Burke[1,2‡], Emily Hyun[1,2‡], Fangyong Li[3‡], Bin Zhou[3‡], John Cahill[1,2‡], Emily R. Kline[4,5‡], Matcheri S. Keshavan[4,5‡], Vinod H. Srihari[1,2]

1 Department of Psychiatry, Yale University School of Medicine, New Haven, Connecticut, United States of America, 2 Program for Specialized Treatment Early in Psychosis (STEP), New Haven, Connecticut, United States of America, 3 Yale Center for Analytical Sciences, Yale School of Public Health, New Haven, Connecticut, United States of America, 4 Department of Psychiatry, Harvard Medical School at Beth Israel Deaconess Medical Center, Boston, Massachusetts, United States of America, 5 Massachusetts Mental Health Center, Boston, Massachusetts, United States of America

☉ These authors contributed equally to this work.
‡ SB, EH, FL, BZ, JC, ERK, and MSK also contributed equally to this work.
* walter.mathis@yale.edu

**Data Availability Statement:** The main content of our data involves patient dates of service, protected health information under US HIPAA laws. Both our IRB and Office of Sponsored Projects agreed that

## Abstract

### Objective

An extensive international literature demonstrates that understanding pathways to care (PTC) is essential for efforts to reduce community Duration of Untreated Psychosis (DUP). However, knowledge from these studies is difficult to translate to new settings. We present a novel approach to characterize and analyze PTC and demonstrate its value for the design and implementation of early detection efforts.

### Methods

Type and date of every encounter, or node, along the PTC were encoded for 156 participants enrolled in the clinic for Specialized Treatment Early in Psychosis (STEP), within the context of an early detection campaign. Marginal-delay, or the portion of overall delay attributable to a specific node, was computed as the number of days between the start dates of contiguous nodes on the PTC. Sources of delay within the network of care were quantified and patient characteristic (sex, age, race, income, insurance, living, education, employment, and function) influences on such delays were analyzed via bivariate and mixed model testing.

### Results

The period from psychosis onset to antipsychotic prescription was significantly longer (52 vs. 20.5 days, [p = 0.004]), involved more interactions (3 vs. 1 nodes, [p<0.001]), and was predominated by encounters with non-clinical nodes while the period from antipsychotic to STEP enrollment was shorter and predominated by clinical nodes. Outpatient programs

we could not share such information without a formal Data Use Agreement. As a compromise, we have added our raw csv – with dates redacted – to our supplementary materials. This gives an essence of how we encoded and computed our data. The points of contact at the Yale Office in Institutional Research is Marta Boeke (marta.boeke@yale.edu) and at Office of Sponsored Projects is Oluma Onuma (uloma.onuma@yale.edu).

**Funding:** None of the authors have any conflicts of interest of financial support to report. This work was supported by National Institutes of Health (R01MH103831) and the Gustavus and Louise Pfeiffer Research Foundation. The funding sources had no role in the design and conduct of the study; collection, management, analysis, and interpretation of the data; preparation, review, or approval of the manuscript; and decision to submit the manuscript for publication. This work was also funded by the State of Connecticut, Department of Mental Health and Addiction Services, but this publication does not express the views of the Department of Mental Health and Addition Services or the State of Connecticut. The views and opinions expressed are those of the authors.

**Competing interests:** The authors have declared that no competing interests exist.

were the greatest contributor of marginal delays on both before antipsychotic prescription (median [IQR] of 36.5 [1.3–132.8] days) and (median [IQR] of 56 [15–210.5] days). Sharper functional declines in the year before enrollment correlated significantly with longer DUP (p<0.001), while those with higher functioning moved significantly faster through nodes (p<0.001). No other associations were found with patient characteristics and PTCs.

## Conclusions

The conceptual model and analytic approach outlined in this study give first episode services tools to measure, analyze, and inform strategies to reduce untreated psychosis.

## Introduction

The Duration of Untreated Psychosis (DUP), or interval between onset of psychosis and initiation of treatment [1], has emerged as an important metric for services targeting recent onset schizophrenia spectrum disorders [2]. Observational studies across varied healthcare systems have confirmed a robust association between prolonged DUP and poorer long-term outcomes [3]. A seminal Scandinavian quasi-experimental test of an early detection (ED) campaign demonstrated that reduction of DUP resulted in a range of positive outcomes, including reduced symptom severity and suicidality at presentation to care, and improved functioning up to 10 years later [4]. However, most ED campaigns have failed to reduce DUP [5, 6]. A challenge inherent to reducing DUP is that it has multi-factorial determinants including patient, illness, family, societal, and treatment system factors [7]. Also, the variety of interacting sets of actors with varying influence across different regions limits extrapolation from one ED campaign to another.

The concept of pathways to care (PTC) [8], has catalyzed a wide range of investigations and revealed myriad factors that impact patients' journeys to and through health systems. As such, knowledge of PTC offers an intuitively compelling way to understand and potentially modify some of the multifactorial determinants of DUP. However, a lack of conceptual clarity and standardized measurement—such as focusing on only certain segments of the PTC or inconsistently defining sub-parts of the PTC—have limited interpretation of findings [9–12]. Also, the often regionally idiosyncratic mixtures of specific determinants have meant that prior reports of PTC can be of little practical value in designing ED for a specific target community [13]. What is called for instead is robust, agnostic PTC collection combined with a formulation and analytic technique that responds to the provincial differences of varied settings.

From an interventionist perspective, early detection efforts need to target local and modifiable sources of delay to impact community level DUP. We conceptualized two broad domains of DUP: a 'Demand' side, that included all factors affecting a patient's help-seeking journey until their psychosis was identified by a healthcare provider able to initiate treatment; and a 'Supply' side that included all factors affecting subsequent delay within the healthcare system until entry into specialty team-based First Episode Psychosis (FEP) services. A further concept was borrowed from microeconomics. Marginal analysis explores the impacts of making a small change to an overall system—e.g., "marginal cost" or the additional cost of producing one more unit [14]. Analogously, *marginal delay* is conceptualized here as the additional delay attributable to a single interaction on a PTC (e.g., how many days a particular visit to the Emergency Department added to DUP).

This paper illustrates a conceptual approach to assessing and analyzing PTCs that was implemented within an active early detection study. We aimed to increase practical utility for future ED efforts by developing a generalizable template that would also permit local adaptation to specific sources of delay. PTCs have been measured and analyzed with an emphasis on: (i) discovering and characterizing sources of delay within a local network of care; (ii) assessing their impact on overall and differential delay for patient subgroups; and (iii) revealing actionable information to refine ongoing ED.

## Methods

### Setting and sampling

The clinic for Specialized Treatment Early in Psychosis (STEP) in New Haven, Connecticut provides an evidence-based model of specialty team-based care for first-episode psychosis to patients aged 16–35, within three years of onset of a schizophrenia-spectrum disorder, and residing in a surrounding 10-town target catchment (population ~400,000, mixed urban and suburban) [15]. Data for this analysis were drawn from a convenience sample of consecutive enrollees (February 1, 2014, to January 31, 2019) during an Early Detection campaign targeting STEP's catchment [13, 16]. All subjects provided informed consent within a protocol approved by the Yale Human Investigations Committee.

### Measures

An adapted version of the Pathways to Care Interview [17] was used to systematically gather information from participants, caregivers, and clinical records about each help-seeking attempt, its symptomatic or behavioral precipitant, to whom participants turned for help, the date, the outcome of the help-seeking attempt, and perceived barriers to accessing care [available upon request]. When the precise date of an interaction could not be recalled, this was approximated to the 1st or 15th of the month.

The date of psychosis onset was operationalized as when Presence of Psychotic Syndrome (POPS) criteria were met on the Structured Interview for Psychosis-Risk Syndromes (SIPS) [18]. Global Assessment of Functioning (GAF) [19] scores, both at the time of enrollment and retrospectively 12 months prior, and Global Functioning: Role and Social [20] scores were also computed during SIPS administration. The SIPS was administered by trained raters who reviewed medical records and interviewed subjects, family members, and referring clinicians. Retrospective scores were based on data collected from SIPS, the screening assessment, and medical records. A medication log included queries for the name, indication, and dose of all antipsychotics prescribed between psychosis onset and enrollment in STEP.

From these intake data, electronic medical records, and patient and family reports we assembled a PTC for each participant, starting with onset of psychosis, proceeding through help-seeking, prescription of first antipsychotic for psychosis, and ending with STEP enrollment (Fig 1). Discrepancies or ambiguities in the data from these structured assessments were reviewed by two trained psychiatrists (WSM, MF). If PTC data defied reconciliation, the participant was excluded from this analysis.

Each participant's PTC was conceived of as a string of interactions, or nodes, with individuals or agencies providing clinical care (e.g. emergency departments, primary care providers, therapists) or those with the capacity to facilitate access to treatment (e.g. family members, police officers, teachers). The former we classified as clinical caregiver nodes, the latter community caregiver nodes. These nodes variably utilized these capacities by hastening (or delaying) access to the local FEP. They thus constituted a *de facto* regional network of stakeholders that STEP could leverage to reduce DUP.

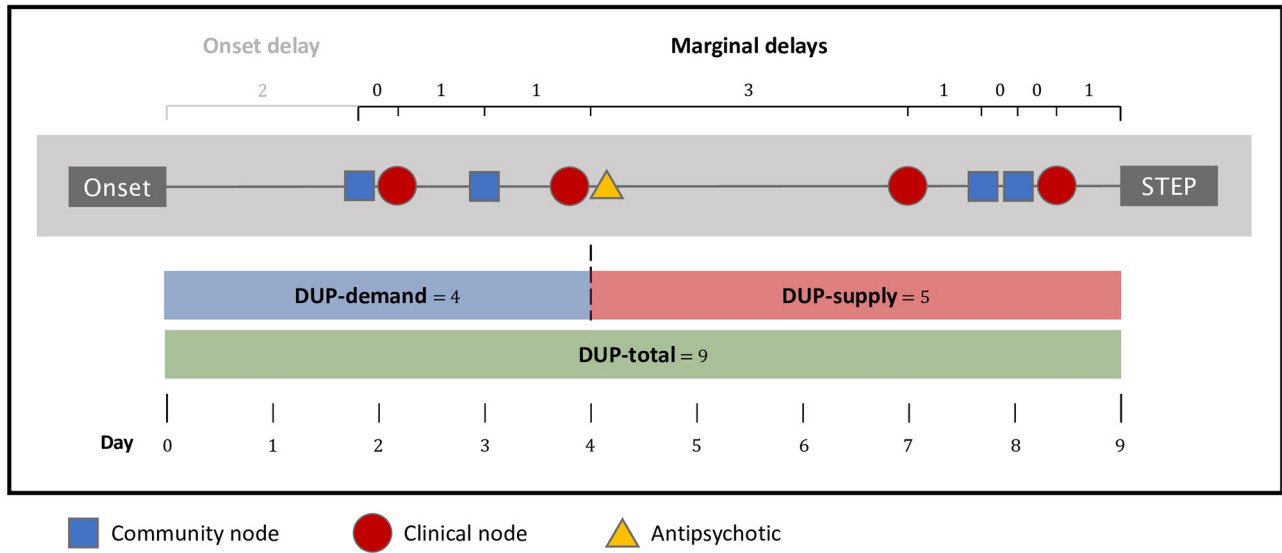

**Fig 1. Conceptual and analytic model of Pathway to Care (PTC) with example values.** This figure depicts the pathway to care described by the following narrative (durations are shorter than sample data for illustrative purposes): Peter began experiencing concerning auditory hallucinations on Day 0 (Onset). Two days later, his mother observes him responding to internal stimuli (first community node) and takes him to an urgent care clinic (first clinical node) the same day. They recommend watchful waiting and outpatient follow up. The next day, Peter is more bothered with the hallucinations and makes an appointment for himself (second community node) at an urgent care psychiatric service the next day. On that visit on Day 4 (second clinical node), Peter is prescribed an antipsychotic to help with his symptoms and an appointment is made for an outpatient psychiatrist, whom he sees 3 days later (third clinical node). On Day 8, Peter's father observes increasingly concerning behaviors (third community node) and takes him to see a youth counselor at their church (fourth community node) who recommends taking him to the local Emergency Department, which they do (fourth clinical node). The ED refers Peter to STEP and he is enrolled the next day, Day 9 (STEP). **DUP-total**: Duration of Untreated Psychosis. **DUP-demand**: Demand-side duration of untreated psychosis, number of days from Onset until antipsychotic. **DUP-supply**: Supply-side duration of untreated psychosis, number of days from antipsychotic until STEP. **Onset**: Onset of psychosis, as ascertained by POPS criteria on the SIPS scale. **STEP**: Enrollment in Specialized Treatment Early in Psychosis clinic. **Onset delay**: Time in days from onset of psychosis first help-seeking node. **Marginal-delay**: Time in days until the next node.

**Global measures.** We computed three global measures of DUP (Fig 1). **DUP-total** was defined as the duration in days from onset of psychosis to enrollment in STEP. This was conceptualized as including delays in successive stages of 'Demand' (i.e., from illness onset to first contact with a healthcare provider who identified and initiated treatment for psychosis) and 'Supply' (i.e., subsequent delays within the healthcare system until engagement with the local FEP or STEP) [13]. **DUP-demand** was operationalized as days from the onset of psychosis to the first administration of an antipsychotic to specifically treat psychosis (e.g., excluding off-label prescriptions for sleep). The latter event was used as a proxy for the first recognition of psychosis by a healthcare provider, and signaling the beginning of **DUP-supply**, which in turn ends with enrollment at STEP. For cases still antipsychotic-naïve at enrollment, DUP-supply was zero (i.e., the entire delay was attributed to Demand side delays).

Initiation of antipsychotic medication is often used in first episode psychosis research to signify that the illness has transitioned into the treatment phase [21, 22]. The prescribing of an antipsychotic for psychosis is a pragmatic and reliable way to index when a healthcare provider, who also has the ability to treat psychosis, has identified psychosis—and hence serves as the transitional event from Demand to Supply.

**Node-level measures.** To better understand how subcategories of nodes differentially affected delay, we computed **marginal-delay** for each node encounter as the time from the

start of each node to the start of the subsequent node (Fig 1). Analyzing these marginal-delays across all nodes on each PTC, and all PTCs within the study, allowed us to examine how DUP was impacted by each node type, and if this varied across participant characteristics.

To assess if global delay measures (DUP-total, DUP-demand, DUP-supply) were influenced by participant characteristics at admission, the following covariates were analyzed: sex, race, age at psychosis onset, reported household income, insurance status, living situation, level of education, employment status, school enrollment status, GAF scores for the month of enrollment (GAF-e), GAF retrospectively assessed for 12 months prior to STEP enrollment (GAF-12), the arithmetic difference between these two (GAF-Δ), Global Functioning: role score (GF-r), and Global Functioning: social score (GF-s). Spearman rank testing was used for continuous independent variables and Kruskal-Wallis testing for categorical independent variables. Given the small sample size, effect size was inferred from the Spearman rho coefficient and computed from the Kruskal H statistics respectively.

An aggregate of all PTCs was used to compute node counts and encounter frequency and marginal-delay for each node type. The contribution to delay by specific node type was described by computing median days of marginal-delay by node type on the Demand and Supply sides across all PTCs.

To understand how participant characteristics influenced delay in transitioning through specific node types, mixed model repeated measures analysis was applied to the participant characteristics with marginal-delay as the outcome measure. Mixed model analyses are useful for analyzing longitudinal data where both fixed and random effects need to be considered, and within- and between-subjects variance can be properly dealt with.

R version 4.0.3 and SAS version 9.4 were used for modelling the data and statistical analyses. A p-value < 0.05 was used to infer statistical significance except in multiple-comparison contexts when a Bonferroni correction was used.

## Results

### Sample

During the study period, there were 1,356 inquiries to STEP, 1,148 were assessed for eligibility, 199 determined eligible, and 171 (85.9%) enrolled. Among enrolled cases, fifteen (8.8%) were excluded from this analysis because of discrepant PTC data. These participants (Table 1) did not differ significantly from the remaining subjects in age, race, or gender (Wilcoxon Rank Sum Test and Pearson's Chi-squared Test respectively). Twelve of the 156 remaining subjects (7.7%) were antipsychotic-naïve at enrollment.

### Network

Aggregating all PTCs identified six community and nine clinical caregiver node types that constituted the regional network of care (Fig 2). The community nodes included family members, police officers, educational representatives such as teachers or counsellors, and nodes that did not fit any other type. The 'Self' community node denotes participants who initiated their own pathway to the next node, without assistance from others. The clinical nodes fell into types reflective of regional services: emergency departments, inpatient psychiatric units, outpatient mental health clinics, intensive outpatient programs, walk-in evaluations in an urgent care outpatient setting, primary care providers, urgent evaluations in walk-in clinics or by mobile outreach teams, mental health clinical care in settings other than those already outlined, and non-mental health clinical care not already outlined.

**Table 1. Demographic and clinical characteristics of participants.** (n = 156).

| | Mean (SD) or Count (%) |
|---|---|
| **Gender, male** | 113 (72.4%) |
| **Race & Ethnicity** | |
| Black, non-Hispanic | 67 (43%) |
| White, non-Hispanic | 40 (26%) |
| Hispanic | 30 (19%) |
| Multi-racial, non-Hispanic | 16 (10%) |
| Other | 3 (5%) |
| **Age at Psychosis Onset** | 21.6 (3.8) |
| **Household Income** | |
| Less than $39,999 | 63 (40%) |
| $40,000 to $59,999 | 25 (16%) |
| $60,000 to $99,999 | 24 (15%) |
| $100,000 and above | 27 (17%) |
| Don't know or refused | 17 (11%) |
| **Insurance** | |
| Public | 84 (54%) |
| Private | 57 (37%) |
| Uninsured | 7 (4%) |
| Other | 2 (1%) |
| **Living Situation** | |
| With Family | 138 (88%) |
| Alone | 7 (4%) |
| With spouse or partner | 2 (1%) |
| Other | 7 (4%) |
| **Highest Level of Education** | |
| Less than high school | 31 (20%) |
| High school | 101 (65%) |
| Technical School/College/University | 24 (15%) |
| **In School Full-time** | 25 (16%) |
| **Employed** | 29 (25%) |
| **GAF**[*] | |
| GAF-e | 31.1 (10.9) |
| GAF-12 | 53.7 (17.5) |
| GAF-$\Delta$ | -22.5 (19.3) |
| GF-r | 4.4 (2.2) |
| GF-s | 5.1 (1.5) |

GAF: Global Assessment of Functioning.

GAF-e: Total GAF at enrollment

GAF-12: Total GAF 12 months prior to enrollment.

GAF-$\Delta$: Difference between GAF-12 and GAF-e

GF-r: Global Functioning: Role Scale

GF-s: Global Functioning: Social Scale

## Clinical Nodes

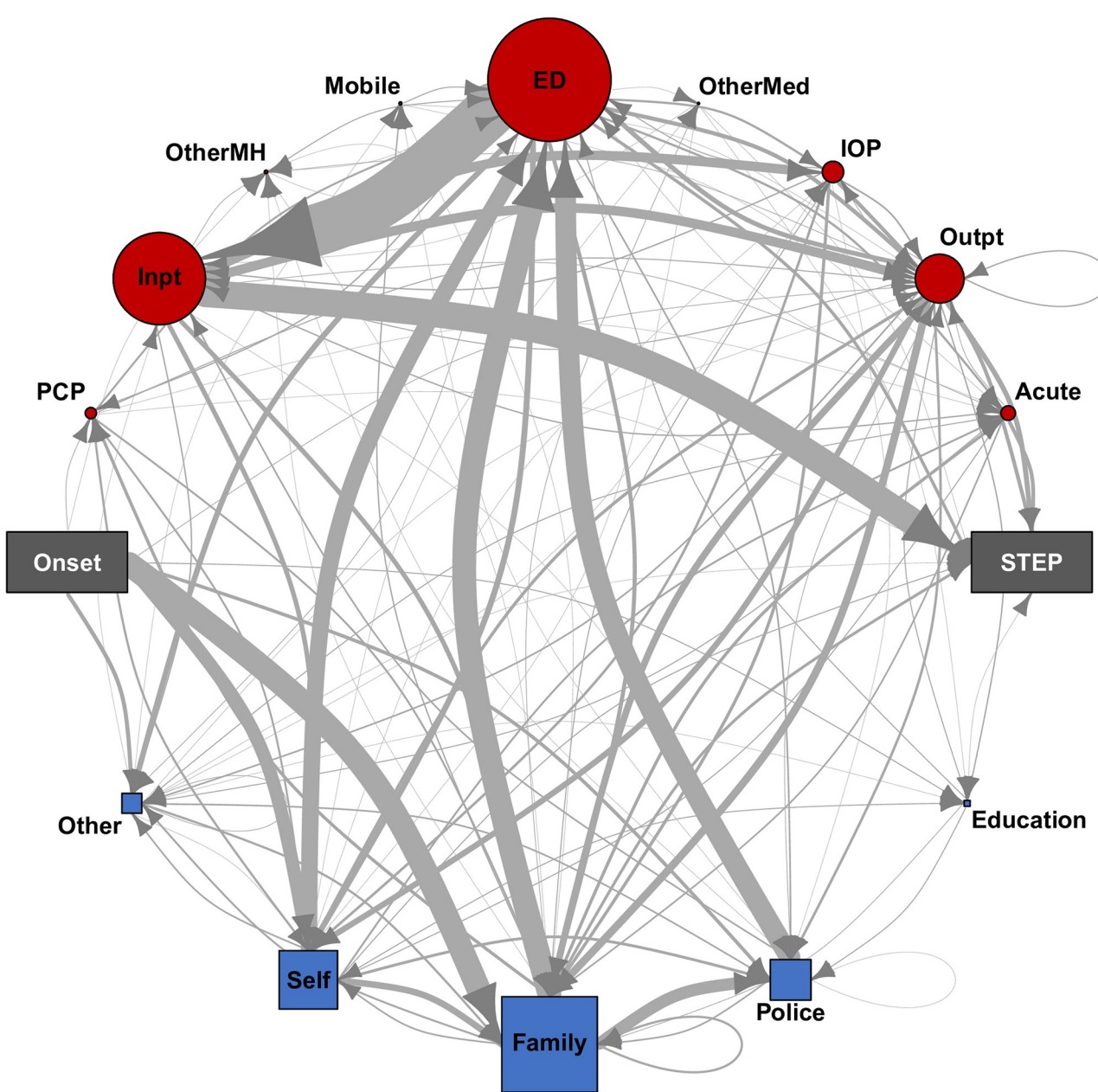

## Community Nodes

**Fig 2. A directional graph of all PTCs collected in this study.** Arrows depict the sequential progression individuals took from Onset to STEP enrollment. Clinical nodes are on the top, community nodes on the bottom. The thickness of the edge (line) between nodes reflects the frequency of traffic between them, and the size of each node reflects the cumulative number of interactions with the node type across all PTCs. Abbreviations: **Self**— self-presented; **Education**—teacher or school counsellor; **Other**—community caregiver not otherwise included; **ED**—Emergency Department; **Inpt**— Inpatient Admission; **Outpt**—Outpatient Mental Health; **IOP**—Intensive Outpatient; **Acute**—Acute Evaluation; **PCP**—Primary Care Provider; **Mobile** —Mobile Evaluation; **OtherMH**—Prison mental health, in-home psychiatric services, substance use disorder inpatient and outpatient; **OtherMed**— Outpatient non-psychiatric, non-PCP (e.g., neurologist), Inpatient medical.

## Measures of overall network performance

Overall, there were more clinical than community nodes (661 vs. 456). Most community nodes (317/456, 70%) were encountered within the Demand side of the PTC while the majority of clinical nodes (433/661, 66%) were encountered within the Supply side—before and after recognition of psychosis by the clinical care network respectively. This is consistent with our formulation of Demand and Supply. A substantial number of clinical nodes (228/661) interacted with FEP patients on Demand side, but we infer did not recognize psychosis. After psychosis onset, it took relatively longer and required more interactions to be prescribed an antipsychotic (DUP-Demand) than to be subsequently referred to STEP (DUP-Supply). Median (IQR) DUP-Demand was more than twice median DUP-supply (52.0 [15–196.2] vs. 20.5 [9–127.8] days, p = 0.004, Wilcoxon rank sum test). On the Demand side each patient encountered a greater number of nodes (median 3 vs. 1 on Supply side, p<0.001, Wilcoxon rank sum test) (Table 2). More total nodes correlated with longer DUP-total and more Supply nodes correlated with longer DUP-supply (both p<0.001), but Demand nodes did not correlate with DUP-demand. Onset delay, or the time from psychosis onset to help-seeking by the patient or others, accounted for 37.5% of DUP-total and 67.4% of DUP-demand.

## Patient factors and delay

Sex, age at psychosis onset, race, household income, insurance status, living situation, educational attainment, school status, and employment status were not significantly associated (and with small effect sizes) with global measures of delay—DUP-total, DUP-demand, or DUP-supply (Table 3). While GAF at enrollment did not correlate with global measures of delay, higher GAF-12 scores correlated significantly with shorter DUP-total, meaning faster access to STEP

**Table 2. Descriptive statistics of node counts and global delay measures delay with correlation testing.**

|  |  | Sum (%) | Median (IQR) |
|---|---|---|---|
| **Node Counts** | **Total** | 1,117 | 5 (4–9) * |
|  | Community | 456 (40.8%) | 2 (1–4) |
|  | Clinical | 661 (59.2%) | 3 (2–6) |
|  | **Demand** | 545 | 3 (2–4) † |
|  | Community | 317 (58.2%) | 2 (1–2) |
|  | Clinical | 228 (41.8%) | 1 (1–2) |
|  | **Supply** | 572 | 1 (1–5) ‡ |
|  | Community | 139 (24.3%) | 0 (0–1) |
|  | Clinical | 433 (75.7%) | 1 (1–3) |
| **Delays** (days) | **DUP-total** | 43,107 | 151.0 (51.8–444.3) * |
|  | **DUP-demand** | 24,019 (55.7%) | 52.0 (15.0–196.2) † |
|  | **DUP-supply** | 19,088 (44.3%) | 20.5 (9.0–127.8) ‡ |
|  | **Onset delay** | 16,178 | 21.5 (2.75–116.8) |

Spearman's rank correlations:

* p < 0.001

† p = 0.15

‡ p < 0.001

**'Demand'**: between psychosis onset and antipsychotic medication prescription

**'Supply'**: between antipsychotic prescription and study enrollment

**'Onset delay'**: time from psychosis onset to first help-seeking node

IQR: interquartile range

**Table 3. Association and effect size testing of patient factors with global delay measures.**

| | DUP-total | DUP-demand | DUP-supply |
|---|---|---|---|
| **Sex*** | p = 0.14 | p = 0.54 | p = 0.40 |
| | $\eta^2 = 0.008$ | $\eta^2 = 0$ | $\eta^2 = 0$ |
| **Age at Psychosis Onset†** | p = 0.33 | p = 0.73 | p = 0.23 |
| | $r_s = -0.08$ | $r_s = -0.03$ | $r_s = -0.10$ |
| **Race or Ethnicity*** | p = 0.35 | p = 0.50 | p = 0.72 |
| | $\eta^2 = 0.003$ | $\eta^2 = 0$ | $\eta^2 = 0$ |
| **Household Income*** | p = 0.20 | p = 0.21 | p = 0.96 |
| | $\eta^2 = 0.01$ | $\eta^2 = 0.01$ | $\eta^2 = 0$ |
| **Insurance*** | p = 0.87 | p = 0.97 | p = 0.76 |
| | $\eta^2 = 0$ | $\eta^2 = 0$ | $\eta^2 = 0$ |
| **Living Situation*** | p = 0.50 | p = 0.99 | p = 0.37 |
| | $\eta^2 = 0$ | $\eta^2 = 0$ | $\eta^2 = 0.001$ |
| **Max Education*** | p = 0.32 | p = 0.19 | p = 0.20 |
| | $\eta^2 = 0.002$ | $\eta^2 = 0.009$ | $\eta^2 = 0.008$ |
| **In School*** | p = 0.94 | p = 0.56 | p = 0.61 |
| | $\eta^2 = 0$ | $\eta^2 = 0$ | $\eta^2 = 0$ |
| **Employed Full-time*** | p = 0.89 | p = 0.80 | p = 0.06 |
| | $\eta^2 = 0$ | $\eta^2 = 0$ | $\eta^2 = 0.02$ |
| **GAF-e†‡** | p = 0.08 | p = 0.23 | p = 0.29 |
| | $r_s = 0.14$ | $r_s = 0.10$ | $r_s = 0.09$ |
| **GAF-12†‡** | **p < 0.001** | p = 0.12 | **p = 0.008** |
| | $r_s = -0.35$ | $r_s = -0.13$ | $r_s = -0.21$ |
| **GAF-Δ†‡** | **p < 0.001** | p = 0.05 | **p = 0.001** |
| | $r_s = 0.43$ | $r_s = 0.16$ | $r_s = 0.27$ |
| **GF-r†‡** | p = 0.67 | p = 0.84 | p = 0.047 |
| | $r_s = 0.03$ | $r_s = -0.02$ | $r_s = -0.16$ |
| **GF-s†‡** | p = 0.89 | p = 0.13 | p = 0.51 |
| | $r_s = 0.01$ | $r_s = -0.12$ | $r_s = 0.05$ |

* Categorical variable analyzed with Kruskal-Wallis testing; $\eta^2$ computed from H-statistic.

† Continuous variable analyzed using Spearman's rank correlation testing

‡ GAF-e: Global Assessment of Functioning, the month before enrollment

GAF-12: GAF 12 months prior to enrollment in clinic.

GAF-Δ: arithmetic difference between GAF-e and GAF-12

GF-r: Global Functioning: Role Score

GF-s: Global Functioning: Social Score

for subjects with higher functioning in the year preceding clinic enrollment. Higher GAF-Δ scores, or greater functional decline in the year before enrollment, correlated significantly with increased DUP-total and DUP-supply. GF-r and GF-s scores did not predict global measures of delay.

Mixed model repeated measures analysis was used to interrogate the impact of patient characteristics on marginal-delay by node type. GAF-12 was an independent predictor with higher scores associated with less marginal-delay (p<0.001) [S1 Table]. Since no other patient characteristic was found to be a significant predictor of marginal-delay, there was no indication to run a subsequent moderation analysis of the association between node type and other variables.

**Table 4. Descriptive statistics of node type encounter frequency and marginal-delay contribution.**

| | | Total Encounters | | Unique participant encounters | | Demand encounters | | Marginal-delay per Demand encounter (days) | | | Supply encounters | | Marginal-delay per Supply encounter (days) | | |
|---|---|---|---|---|---|---|---|---|---|---|---|---|---|---|---|
| | | (n = 1,117) | | (n = 156) | | | | Median | IQR | Range | | | Median | IQR | Range |
| **Community** | **Family** | 198 | 17.7% | 121 | 77.6% | 140 | 44.2% | 0 | 0–0 | 0–519 | 58 | 42% | 0 | 0–0 | 0–126 |
| | **Self** | 121 | 10.8% | 62 | 40% | 72 | 23% | 0 | 0–0 | 0–61 | 49 | 35% | 0 | 0–0 | 0–14 |
| | **Police** | 84 | 7.5% | 63 | 40% | 63 | 20% | 0 | 0–0 | 0–149 | 21 | 15% | 0 | 0–0 | 0–30 |
| | **Other** | 41 | 3.7% | 31 | 20% | 32 | 10% | 0 | 0–0 | 0–388 | 9 | 6% | 0 | 0–0 | 0–24 |
| | **Education** | 12 | 1.1% | 10 | 6.4% | 10 | 3.2% | 0 | 0–0 | 0–954 | 2 | 1% | 8.5 | 4.3–12.8 | 0–17 |
| | **Total** | 456 | 40.8% | | | 317 | | | | | 139 | | | | |
| **Clinical** | **ED** | 255 | 22.8% | 137 | 87.8% | 131 | 57.5% | 0 | 0–1 | 0–69 | 124 | 28.6% | 0 | 0–2 | 0–187 |
| | **Inpt** | 191 | 17.1% | 122 | 78.2% | 25 | 11% | 12 | 9–18 | 3–335 | 166 | 38.3% | 13 | 9–21 | 0–820 |
| | **Outpt** | 101 | 9.0% | 65 | 42% | 30 | 13% | 36.5 | 1.3–132.8 | 0–584 | 71 | 16% | 56 | 15–210.5 | 0–724 |
| | **IOP** | 44 | 3.9% | 28 | 18% | 2 | 0.9% | 19.5 | 9.8–29.3 | 0–39 | 42 | 9.7% | 29 | 17.8–46 | 1–896 |
| | **Acute** | 30 | 2.7% | 26 | 17% | 14 | 6.1% | 5.5 | 2.3–19.5 | 0–305 | 16 | 3.7% | 5.5 | 0.75–24.5 | 0–333 |
| | **PCP** | 23 | 2.1% | 19 | 12% | 17 | 7.5% | 2 | 0–6 | 0–354 | 6 | 1% | 1.5 | 0.25–46.3 | 0–129 |
| | **OtherMH** | 7 | 0.6% | 4 | 3% | 3 | 1% | 0 | 0–0 | 0–0 | 4 | 1% | 107.5 | 28.5–191 | 21–212 |
| | **Mobile** | 6 | 0.5% | 6 | 4% | 5 | 2% | 0 | 0–1 | 0–1 | 1 | 0.2% | 0 | - - | - - |
| | **OtherMed** | 4 | 0.4% | 4 | 3% | 1 | 0.4% | 109 | - - | - - | 3 | 1% | 41 | 24.5–95.5 | 8–150 |
| | **Total** | 661 | 59.2% | | | 228 | | | | | 433 | | | | |

### Node categories

Family was the most frequently utilized community node, both in total number of encounters (198/456, 43.4%) and number of participants (121/156, 77.6%) (Fig 2 and Table 4). Interactions with police and self-referral were also common (40% of participants in each category). While, as noted, community node encounters skewed toward the Demand side (317/456), the distribution of community node types changed between the Demand and Supply time periods, with 'self' occurring relatively more frequently, and the other node types occurring less frequently on the Supply side. Community nodes had little marginal-delay.

Emergency department was the clinical node type encountered most frequently (255/661, 38.6%) and by the largest number of participants (137/156, 87.8%). Moreover, it was encountered repeatedly (median [IQR] of 1 [1–2] encounters per participant with ED encounter), as were inpatient psychiatric units (median [IQR] of 1 [1–2] admissions per participant with inpatient admission). Emergency departments were a larger percentage of Demand side clinical nodes than Supply side (57.5% vs 28.6%) while the reverse was true for inpatient admissions (11% vs 38.3%). Among highly utilized clinical nodes, outpatient mental health was the biggest contributor of marginal-delay on both Demand (median [IQR] of 36.5 [1.3–132.8] days) and Supply sides (median [IQR] of 56 [15–210.5] days).

### Discussion

This study aimed to conceptualize, measure, and analyze PTCs in a manner that can provide actionable information for ED initiatives across diverse settings. Recognizing that DUP is an important global measure with many contributors, other ED efforts have devised approaches for subdividing delay by partitioning the PTC by key events [23]. But our more granular approach can reveal node-specific regional patterns and suggest interventions for either

immediate implementation within a performance improvement framework, or research to develop novel approaches for refractory sources of delay.

## Global delay

Onset delay, or the time from psychosis onset to help-seeking, was comparable to previous work using similar methods [24] and a large contributor to DUP which might help explain why so many PTCs went through ED and inpatient settings and why Demand nodes did not add much to delay (i.e., help-seeking may be initiated so long after psychosis begins that symptom acuity can only be safely managed in an ED or inpatient setting). These settings are not always conducive to FEP engagement and our data show multiple interrupted healthcare contacts on the Supply side. This validates the utility of multi-component ED campaigns [4, 13] that can target both willing patients and, when this is not the case, those around them who could facilitate their access to care. As frequently utilized community nodes, families, and police are logical targets for specific messaging in our region.

Onset delay is a difficult metric to reduce because it predates interactions with the mental health system and contributions to this delay are multifactorial, including insight into early symptomatology, healthcare access disparities and previous aversive experiences with healthcare, individual and family views of mental health treatment and stigmatized views of mental illness [7, 25]. As such, this suggests that ED campaigns need to incorporate lay-facing psychoeducation campaigns to help identify early symptoms and recommend first steps toward care [13].

We had previously reported more self-initiation and shorter DUP-demand amongst those who sought help during the prodromal illness phase when insight is relatively preserved [26]. In this study, individuals, once identified as having psychosis, were shuffled more quickly through predominantly clinical caregivers, but also appeared to play a larger role in initiating these encounters themselves, albeit suffering multiple interrupted healthcare interactions *en route* to STEP. Possible mechanisms for this include interactions between participant or illness factors (e.g., increased symptom severity interfering with treatment engagement, or increased experience with the healthcare system enabling more help-seeking), treatment system factors (e.g., challenges in transitioning care from inpatient to outpatient settings), and care experiences that can be aversive (e.g., criminal justice interactions or involuntary hospitalization).

## Participant factors

Our finding that sex, race, insurance status, employment status, GAF at enrollment do not correlate with global measures of delay aligns with similar studies in the US [12, 21]. But those studies did find correlations with age, living situation, education level, and school status that we did not. These discrepant findings possibly result from differences in cohort characteristics or methods of tabulation (e.g., grouping ages instead of treating as continuous). There are methodological limitations that limit the strength of these conclusions, especially around race [27], such that larger sample sizes within and across regions are necessary to fully interrogate disparities in access.

Those with better functioning 12 months prior to enrollment and, even more so, those with greater functional decline during that period, had significantly lower DUP. The first observation is not surprising in that higher functioning participants may be better able to navigate local pathways. The second suggests a salutary interaction between a greater need for care resulting in a quicker response from the healthcare system. Both features of the network can inform future outreach efforts to community and clinical nodes.

## Node type and marginal-delay

There was high variability of marginal-delay from clinical nodes, exemplified by the difference in delay from emergency departments (short, often admitting quickly to inpatient units) and the long delays from outpatient mental health. This result calls for direct outreach to outpatient providers to both recommend referral to FEP and better understand factors that have slowed referrals.

The high utilization of emergency departments and inpatient units both highlights the importance of working with these local care centers to facilitate connection with FEP when clinically appropriate and speaks to the symptomatic acuity of the participants. However, resources that are designed to more proactively manage high acuity in the community (e.g., urgent care outpatient evaluations or mobile evaluations) were not highly utilized. This suggests that the availability and/or awareness of these resources was less than that of the emergency departments, as reported for other regions [28]. Furthermore, reducing the proportion of individuals who require urgent/emergent care before starting their treatment at an FEP could improve the experience of entering care, minimizing unnecessary aversive experiences that might interfere with future treatment engagement.

Contrary to reports, mainly from the UK, that primary care providers (PCPs) were the most common first PTC contact [29], and/or the main source of referrals to FEP [30], in our sample only 12% of participants attributed PCP involvement in their PTC, despite 25/30(83%) consecutive participants surveyed (for internal audit purposes) reporting visiting a PCP within the two years prior to enrollment. This finding points to PCPs as a focus for improvement in clinical node outreach, with lessons available from other countries [31, 32].

## Strengths and limitations

The approach outlined above improved upon previous PTC analyses in two domains. The absence of common terminology and metrics makes comparing and synthesizing DUP research difficult. Some studies have focused on the impact of DUP from key parts of the PTC such as number of clinical nodes or type of first clinical interaction [12, 33], while others have subdivided the PTC by threshold events (e.g., first mental health worker, first antipsychotic, or FEP enrollment) to define transition between phases of PTC (eg "help-seeking" and "referral pathways") [24, 25].

But, by being inclusive and agnostic about the relative importance of particular nodes when collecting PTCs, we were able to construct a model of the actual local network of care—both community and clinical—that brought enrollees to our FEP, and discover gaps in expected participation (e.g., by PCPs). Quantitatively, this also permits us a more nuanced understanding of the relationship between DUPs and node counts. Some of these findings were intuitive—such as more total nodes correlating with longer DUP (an analogue of which was also found in Marino, et al. [12])—or less so—such as the number of Demand side nodes not significantly correlating with DUP-demand. Further, the shift of community nodes dominating Demand side to clinical nodes dominating Supply side not only informs intervention strategies but is a finding that previous analyses could not produce.

While we feel that antipsychotic prescription is a pragmatic and reliable liminal event between Demand and Supply, there are conceivable situations where it might not appropriately detect this transition. For instance, given the young age of our sample, one could imagine a prescriber recognizing psychosis in a patient but being reluctant to prescribe an antipsychotic or the patient or patient's family being uninterested in taking it—either concern over side effects, stigma, or perhaps feeling they could manage their current symptoms without

pharmacological intervention. In either case, our model would miss a conceptual transition to Supply. But it would be impossible to infer such thinking from prescribers in our retrospective assessment and we did not ask participants about these more nuanced points.

Marginal-delay analysis is the other unique aspect of this study which we believe brings important quantitative findings essentially undiscoverable in previous work. Marginal-delay allows us to model the impact on delay at the more granular level of node type which is especially important when formulating delay-reducing interventions, where specificity allows tailoring of messaging as well as better allocation of limited financial and logistical resources.

A strength of this study is the collection of granular PTC data from multiple sources, circumventing the assumptions and data limitations of studies analyzing extractions from singular electronic medical records. While we incorporated data from many sources, reporting from participants and their families played a major part in reconstructing PTCs. As such, our data is vulnerable to recall bias to the exact timing of events and variability of which events are more likely (e.g. emergency room visit) or less likely recalled. Even with multiple sources, 8.8% of participants were excluded because their full PTCs could not be reliably reconstructed. Also, the PTC data collected and analyzed were limited to those enrolled in the STEP program. Only a small percentage of those referred to STEP were deemed appropriate for enrollment and the true FEP incidence in the STEP catchment is likely larger than that enrolled in STEP. We do now know how the PTC and delay data of these groups compare to our sample and should be careful when generalizing our findings.

A potential limitation to this study is the treatment of all node types as internally homogeneous (e.g, all Outpatient providers as one group). This is an approach similar to other studies when treating interaction types categorically [12]. We felt it struck an appropriate balance of granularity between overly broad categorization of "healthcare provider" and the overly specific individual provider yet still lends real world applicability to findings by distinguishing coherent targets for specific messaging and outreach efforts within a local network (e.g., emergency departments, educational counsellors, or even primary care providers).

This method of analysis focuses heavily on the systems-related factors and less on other individual and family factors that influence DUP and PTCs [7]. Evidence suggests that individual, family, and stigma factors impact help seeking, and the quality of experiences with mental health services expedite or delay entry into specialty care [25]. As such, further analysis is warranted to better understand how patient and family factors influence navigation along PTCs. This would extend the current literature on attitudinal barriers to help seeking [25, 34, 35]. Also of interest is how community caregivers moderate delay by influencing which clinical caregivers are visited, as well as how participant factors influence referral/routing decisions. Further, this approach can be combined with other assessments of network performance such as spatial analysis [36].

In keeping with our conceptualization of how to quantify and analyze nodes and delays, the particular factors found to be associated with increased delays may not generalize to other settings or even persist over time in our catchment. Rather, our methodological approach offers a template for FEP to implement ED in a manner that is ecologically inclusive, responsive to the local network, and can support performance improvement. We hope to encourage the collection of PTC data from research and clinical first episode psychosis settings. The approach outlined can be replicated in other systems, providing a progressively detailed map of PTC to CSCs across the US and beyond. Comparisons across clinics within this common framework would allow for sharing of lessons and performance improvement, consistent with the concept of learning healthcare systems [37].

## Conclusion

A considerable body of evidence demonstrates the negative impact of lengthy DUP, and great effort is being put into reducing it via ED initiatives. The robust PTC data collection, conceptual model, and analytic approach outlined in this study give first episode services specific, actionable insights on how to best measure, analyze, and focus ED efforts as well as provide a tool for further research on DUP reduction strategies.

## Supporting information

**S1 Table. Mixed model repeated measure analysis of effect of patient characteristics on marginal-delay as outcome measure.** The unadjusted model tests for correlation of individual predictors with the dependent variable marginal-delay. The adjusted or multivariate model test for correlation of each predictor when other predictors occur.
(DOCX)

**S1 Data. Raw pathway to care data.** Dates of services are redacted for confidentiality.
(CSV)

## Author Contributions

**Conceptualization:** Walter S. Mathis, Maria Ferrara, Emily Hyun, John Cahill, Matcheri S. Keshavan, Vinod H. Srihari.

**Data curation:** Walter S. Mathis, Maria Ferrara, Shadie Burke, Emily Hyun.

**Formal analysis:** Walter S. Mathis, Fangyong Li, Bin Zhou.

**Funding acquisition:** Vinod H. Srihari.

**Investigation:** Walter S. Mathis, Vinod H. Srihari.

**Methodology:** Walter S. Mathis, Maria Ferrara, Fangyong Li, John Cahill, Emily R. Kline, Matcheri S. Keshavan, Vinod H. Srihari.

**Project administration:** Walter S. Mathis, Vinod H. Srihari.

**Software:** Walter S. Mathis.

**Validation:** Walter S. Mathis, Maria Ferrara.

**Visualization:** Walter S. Mathis.

**Writing – original draft:** Walter S. Mathis, Vinod H. Srihari.

**Writing – review & editing:** Walter S. Mathis, Maria Ferrara, Shadie Burke, Emily Hyun, Fangyong Li, Bin Zhou, John Cahill, Emily R. Kline, Matcheri S. Keshavan, Vinod H. Srihari.

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
