## [Decision Letter · Decision Letter 0]

9 Feb 2022

PONE-D-21-32200Granular analysis of pathways to care and durations of untreated psychosis: A marginal delay modelPLOS ONE

Dear Dr. Mathis,

Thank you for submitting your manuscript to PLOS ONE. After careful consideration, we feel that it has merit but does not fully meet PLOS ONE’s publication criteria as it currently stands. Therefore, we invite you to submit a revised version of the manuscript that addresses the points raised during the review process.

We look forward to receiving your revised manuscript.

Kind regards,

Giuseppe Carrà, PhD

Academic Editor

PLOS ONE

Journal Requirements:

Reviewers' comments:

Reviewer's Responses to Questions

**Comments to the Author**

1. Is the manuscript technically sound, and do the data support the conclusions?

Reviewer #1: Yes

Reviewer #2: Partly

2. Has the statistical analysis been performed appropriately and rigorously? 

Reviewer #1: Yes

Reviewer #2: Yes

3. Have the authors made all data underlying the findings in their manuscript fully available?

Reviewer #1: No

Reviewer #2: No

4. Is the manuscript presented in an intelligible fashion and written in standard English?

Reviewer #1: Yes

Reviewer #2: Yes

5. Review Comments to the Author

Reviewer #1: Strengths of the paper: Well justified background. Data from a well-established and methodologically sound programme for first episode psychosis. Conceptual model is very well developed. The network node analysis is particularly interesting and novel – it certainly adds value to the field as it indicates likely nodes that both affect PTC and delay. The functioning results are also very interesting.

The issues below require addressing in a revised version – for the most part these should be straightforward to do. My most substantive comment is the last – I think there needs to be some substantive reworking of the discussion to give credence to some possible limitations of the work. Other areas of the discussion could be shortened (particularly the initial summary) to keep the good flow and precision which this work otherwise exhibits.

1. In general, take care throughout in using the term “significant”. It should generally be reserved to denote a statistically significant result (with test statistics reported), with another word used to describe other important findings. i.e. P12 “However, a significant number of Clinical nodes interacted with FEP patients but did not appear to recognize psychosis.” Here this is interesting, but perhaps “substantial” is a more appropriate qualifier – make sure to back up such claims with the actual data.

2. Please report IQR alongside any medians reported in the paper (including abstract).

3. Please report the type of statistical tests used when reporting p-values.

4. In abstract, first sentence of results could be deleted, or values that are reported in second sentence could be integrated into first sentence, to shorten. Report how much longer (in days) the DUP was for those with sharper functional decline.

5. P5, preference to use First Episode Psychosis [FEP] services as the nomenclature to distinguish this type of specialism from first episode programmes for other disorders, and more consistently used in the (at least European) literature.

6. P6, methods: When was the GAF measured on participants? This is stated on p9, but I recommend moving it to where this is first introduced. More detail of how GAF 12 months prior to STEP enrolment should be given.

7. P8, methods: I recommend removing “age at STEP enrolment” from the analyses and paper. This cannot influence delay, and any associations here will be uninformative because they will be perfectly correlated with age at symptom onset and DUP.

8. P9, please report the corresponding statistical test and p-values for the statement “These participants did not differ significantly from the remaining subjects in age, race, or gender.”

9. P13, Table 2 – it’s unclear from the table whether the correlations are for DUP-total to total nodes, DUP-demand to Demand nodes, DUP-supply to Supply nodes. I assume this is the case, but this should be clarified in the table, and text on P12.

10. P13, Table 3: Please retitle the table to avoid the use of the word “correlations” – technically I think only the continuous-continuous associations are correlations; associations is preferred.

11. P16: For consistency it is probably easier to report the median ED encounters per participant than the mean number (with IQR).

12. Discussion: repetition of results (i.e. values) should be avoided in the discussion.

13. Discussion: a separate “strengths and limitations” section should be provided with a subheading. In particular, the paper should discuss:

a. The extent to which data collected in their study (i.e. the reconstruction of PTC) was reliant on clinical records versus patient and caregiver reports. The former are collected prospectively, and thus minimise recall, whereas the latter may be affected by recall, which may act differentially – particularly by level of functioning which could have the potential to bias the results. If the authors have data on the proportions of PTC records rated from these sources, this would be helpful additional information; if this was not recorded, a more detailed discussion on this general point will suffice.

b. The validity of defining DUP-supply as the date from the initiation of antipsychotics. It is possible for clinical contacts before this point to “fail” to prescribe Aps, which could (retrospectively) be seen as a supply-side delay in treatment. Alternatively, is a clinical contact (without Aps) a possible treatment in itself – i.e. if they received psychological interventions at this point or some other form of active monitoring?

c. Detection bias – the sample is, by definition, restricted to those who are referred to and enrolled in STEP. There may be other FEP cases in the community (i.e. the true incidence) that have not yet been referred to STEP. Although the extent of this is unknown, a comment on it is worthwhile – in the true population with FEP in the STEP catchment, could it be that the true DUP was even longer than observed here.

d. Given the high proportion of “false positive” referrals to STEP (i.e. 199/1356 enrolled) could STEP be a significant supply-side node contributing to marginal delays for people with other (non-psychotic) psychopathologies that require onward referral to other specialised mental health programmes (where available)?

e. Could the absence of variation in delays by demographic characteristics be a function of statistical power and the small sample size? Table 3 only reports P-values, but could more helpfully report effect sizes / differences between the comparator groups.

Reviewer #2: The authors have developed an approach to analyze delays in the pathways to care for patients with psychosis. They developed and tested this method on data from 156 participants enrolled in an early treatment program. This proof of concept on a specific sample provides insight into the associations between certain events and delayed care, and is an interesting approach to quantify and analyze pathways to care. The manuscript could benefit from clarifying the assumptions and approach so that the importance become more clear.

Abstract:

From the information in the abstract, it is not immediately clear what “Community nodes” are. The abstract should be interpretable without having to read the manuscript. After reading the manuscript, I think it could be clarified partially by defining the antipsychotic prescription as the switch from demand to supply.

“referred to the clinic” -> does this mean STEP?

Choose “v” or “vs” when comparing two items, not both.

It seems like the words community, clinical, outpatient programs don’t need to be capitalized, as they’re used as-is in this context, and it makes it harder to read. Later on in the manuscript, the same applies to marginal delay. I’m still on the fence regarding Demand and Supply, but see comments in introduction and methods on using these terms in general. Regardless, without defining Demand and Supply, these terms are currently meaningless in the abstract.

“Other associations” should be defined here or in the methods section of the abstract.

The manuscript only provides “actionable insights on how to focus efforts on strategies to reduce untreated psychosis”, but not “how to best measure or analyze strategies”; Only one approach is analyzed in the manuscript and so there is no head-to-head analytical (nor descriptive) comparison of this approach with other methods.

Furthermore, what are the actionable insights? Please state that instead of just mentioning that there are actionable insights.

Introduction:

The paragraph “The concept of pathways…” hints at prior studies that dissect pathways to care (beyond just measuring the DUP), but some specifics would be good to have here. It is not immediately clear that this study will be any different from the prior literature; why would it not have the same shortcomings? In this paragraph, the most recent citation is from 2014 – has there been no more efforts in this field after this year?

The PTC of patients is complex, multifaceted and heterogeneous, and I commend the authors for attempting to create a general measure that captures all aspects. The assumptions of the approach, however, could use some clarification. The following paragraph “From an interventionist perspective…” is not defining the treatment. Later it becomes clear that this treatment event only consists of being prescribed antipsychotic prescription drugs. Why exactly is anti-psychotic prescription the defining moment where it switches from demand to supply? As DUP could be defined in multiple ways, the authors may wish to expand this section to provide the underpinnings of what is considered the end of the DUP (i.e. why would only STEP and not any other treatments be considered the start of the treatment). What if the patient receives outpatient care from a psychiatrist (with or without medication) before being enrolled in STEP, wouldn’t this be considered the end of the untreated period, especially if STEP wasn’t an option? Some of these details may be better suited for the methods section, but overall, more justification is needed for the general approach, and why it is better than everything else that has been published before. If the authors’ main purpose is provide a proof of concept, and not derive any conclusions from the content because of these assumptions, this should be more clearly mentioned.

Methods:

What is the size and type of catchment area of STEP, is it rural, urban, suburban etc?

Were there any differences in recall rates between the place of care? E.g. an emergency room visit is more memorable than perhaps mentioning symptoms at a yearly physical exam.

Enrolment should be enrollment (also in legend of Table 1, and discussion).

From the examples in the legend of figure 1, it becomes somewhat clear what kind of agents define a community node and clinical node, but these terms should be explained in the text of the methods with a comprehensive definition (or a list in the supplement of all the possible types of events that were mentioned by the patients).

Furthermore, the authors later on make a distinction between community and clinical nodes that, later on, appears to be a critical concept in the analysis. The importance should be described in the introduction.

Related to a comment on the introduction - what happens when the patient does not get, want, need, or take anti-psychotics? Especially considering the younger age of the sample, anti-psychotics may not be a desirable intervention for all patients. If they get only a one-month fill of anti-psychotics during the DUP period, and don’t get a refill, according to the current definition, shouldn’t they switch back from supply to demand?

Did the patients provide information on when they had an unmet need of care, or whether symptoms were manageable without care during the DUP period? Would it be possible to inquire with the participants to find out at what stages they considered to be in the demand/supply period? I am not necessarily dismissing the assumptions that are admittedly hard to define and generalize among all patients, but it appears that the authors are making generalized decisions about the patients’ needs without their direct input. With the current level of detail it is hard to determine whether that is a reasonable approach to the problem. I am now aware that other papers describing STEP have used the demand/supply terms, but it appears that these words were defined differently: ‘Demand’: identification of illness and initiation of help-seeking; and ‘Supply’: correct identification of diagnosis and referral to first-episode services (from Srihari et al, BMC Psychiatry 2014). Perhaps a better choice of words in this context would be pre-medication/post-medication. This would also make it clear to the readers (who may not read the methods carefully) that that event is defined as the switch. Or a different approach would be to follow the original definition.

Figure 1 shows a very short DUP, considering that the median DUP in the study sample was 151 days. What was the range? It would be more representative to show a longer DUP in the figure, or at the very least acknowledge in the legend that the duration is short for illustrative purposes. If allowed for privacy reasons, I think it would be interesting if the timelines for all or a representative subset of patients could be shown in a figure (probably in a supplemental section), since there are only 156 patients. This would give a better idea of the distribution of DUP time, and the sequences and kind of nodes that occurred in the whole sample. For researchers who wish to apply the approach developed by the authors, this may give a better idea of whether it would be an informative and feasible approach for their own group of patients, especially considering the general heterogeneity of patient samples and enrollment in a specialized treatment program in this study.

A delay is described as the days after the event until the next. However, the counterfactual is not known. A community node may slow down or speed up the next event if the event had not occurred, but this is unobserved. How do the authors reconcile not knowing the counterfactual?

A few more details on the mixed model methods would be helpful.

The authors would be encouraged to share their R and SAS code used for the analyses to enable other researchers to apply their approach.

Results

The word note in the “Network” section should be node.

The results in the sentence “Unsurprisingly, the majority…” are not surprising because they’re in part mechanical. It might be good to add this, for those who did not read the methods carefully. Similarly, the sentence “While community node encounters skewed toward the demand side” later on in the Node category section, may be clarified.

In the sentence “However, a significant number of clinical nodes…” it would be good to quantify this in some way.

The sentence “Interactions with police…” makes it initially seem like the authors combined police and self-referral while these are very different from each other. After inspecting the table, it appears that it’s just the percentages that are the same. This could be clarified with writing “(40% of participants in each category)” or something similar.

“… the relative frequency of encounters per node type…” doesn’t appear stable -as mentioned to a certain extent- as the self category is higher, and all the other categories are lower. Perhaps the sentence should read: The distribution of the community node types changed between the demand and supply time period, with ‘self’ occurring relatively more frequently, and the other node types occurring less frequently on the supply side.

In general, delays seem to be causally attributed to nodes in this manuscript, but causality cannot be inferred from these methods . Throughout the text the wording (e.g. “contribute” to delay, or making the jump to delay-reducing interventions) should be changed to reflect that

Discussion:

The results don’t necessarily “illustrate robustness of the approach”. I didn’t see any sensitivity analyses, which is acceptable for the purposes of this manuscript, but this sentence should probably be left out. The results also don’t “validate the analytical approach”. The results are very interesting and informative from an explorative viewpoint, but the authors should be careful not to overstate the conclusions and implications of the study. First and foremost, more research is needed before it is clear that this method would be equally informative with other patients or in other contexts (e.g. no enrollment in the STEP program). An explanation of why the results may not generalize would be helpful.

The structure of the sentence “This validates the utility of multi-component ED campaigns…” could be improved.

A sensitivity analysis that includes the partially known PTCs of currently excluded participants may be informative, as in other contexts it is likely not clear whether PTCs are complete or not.

Conclusion

See comments regarding the conclusion in the abstract section.

6. PLOS authors have the option to publish the peer review history of their article (what does this mean?). If published, this will include your full peer review and any attached files.

Reviewer #1: **Yes: **Prof James B. Kirkbride

Reviewer #2: No

---

## [Author Response · Author response to Decision Letter 0]

20 May 2022

We appreciate the close reading and thoughtful comments by editorial staff and both reviewers. Below you will find an item-by-item response to each recommendation.

---

## [Editor Report · Decision Letter 1]

7 Jun 2022

Granular analysis of pathways to care and durations of untreated psychosis: A marginal delay model

PONE-D-21-32200R1

Dear Dr. Mathis,

We’re pleased to inform you that your manuscript has been judged scientifically suitable for publication and will be formally accepted for publication once it meets all outstanding technical requirements.

Kind regards,

Giuseppe Carrà, PhD

Academic Editor

PLOS ONE

---

## [Editor Report · Acceptance letter]

21 Jun 2022

PONE-D-21-32200R1 

Granular analysis of pathways to care and durations of untreated psychosis: A marginal delay model 

Dear Dr. Mathis:

I'm pleased to inform you that your manuscript has been deemed suitable for publication in PLOS ONE. Congratulations! Your manuscript is now with our production department. 

Kind regards, 

on behalf of

Dr. Giuseppe Carrà 

Academic Editor

PLOS ONE